Rare evidence for ‘gnawing-like’ behavior in a small-bodied theropod dinosaur

Brown Caleb M. caleb.brown@mail.utoronto.ca 1
Tanke Darren H. 1
Hone David W.E. 2
1 Royal Tyrrell Museum of Palaeontology , Drumheller , Alberta , Canada
2 School of Biological and Chemical Sciences, Queen Mary University of London , London , United Kingdom
Knoll Fabien
Electronic publication date: 2021 Jun 23
Publication date: 2021
Volume: 9
Electronic Location ID: e11557
Received 2021 Feb 5; Accepted 2021 May 12
Copyright: ©2021 Brown et al.
Copyright year: 2021
Copyright holder: Brown et al.
License: This is an open access article distributed under the terms of the Creative Commons Attribution License, which permits unrestricted use, distribution, reproduction and adaptation in any medium and for any purpose provided that it is properly attributed. For attribution, the original author(s), title, publication source (PeerJ) and either DOI or URL of the article must be cited.
License URL: https://creativecommons.org/licenses/by/4.0/

Keywords: Tooth mark, Gnawing, Ungual, Carcass, Bite mark, Osteophagy, Feeding, Hadrosauridae, Theropoda, Dinosauria

Funding: The Royal Tyrrell Museum of Palaeontology (Government of Alberta) The work was supported by the Royal Tyrrell Museum of Palaeontology (Government of Alberta). The funders had no role in study design, data collection and analysis, decision to publish, or preparation of the manuscript.

==============================
Mammalian carnivores show a higher degree of prey bone utilization relative to non-avian theropod dinosaurs, with this major ecological difference reflected in the frequency and morphology of tooth marks in modern and Cenozoic assemblages relative to Mesozoic ones. As such, prey bone utilization (i.e., gnawing, bone-breaking, osteophagy) may represent a key ecological strategy repeatedly exploited by mammalian carnivores but rarely in theropod dinosaurs. Here we describe an isolated adult-sized hadrosaurid pedal ungual (III-4) from the Dinosaur Park Formation (Campanian) of southern Alberta which shows a unique pattern of bite marks from a small- to medium-sized theropod dinosaur. Thirteen distinct tooth marks occur in a restricted area of the ungual, and the pattern suggests up to six repeated, high-power bites delivered to the bone. The tracemaker cannot be definitively identified, but was likely a dromaeosaurid or very young tyrannosaurid. Tooth marks on at least four other Dinosaur Park Formation hadrosaur pedal unguals are reported, but the overall frequency of occurrence in unguals (< 1%) is much lower than that reported for other bones. The pattern of tooth marks on this specimen deviates from most described theropods tooth marks, and given the low volume of meat associated with the ungual, may represent theropod prey bone utilization as part of late-stage carcass consumption, and a behavior similar to mammalian gnawing.

Introduction

A major ecological and feeding behavior distinction between Mesozoic non-avian theropod and modern and Cenozoic mammalian carnivores is the difference in utilization of prey bones as a food source (Fiorillo, 1991). Both modern and fossil carnivorous mammalian species have been shown to make extensive use of prey bones as a dietary source (Kruuk, 1972; Haynes, 1980; Van Valkenburgh, 1996). This is often characterized by repeated, high-power bites to bone extremities by premolars and molars, often for the purpose of exposing the lipid and nutrient rich marrow (Van Valkenburgh, 1996; Van Valkenburgh, 2007). This behavior may be considered ‘gnawing’, in that it follows the definition ‘to bite or chew something repeatedly, usually making a hole in it or gradually destroying it’ (Cambridge Dictionary, 2016). This gnawing behavior is also often taxon and season specific, allowing for ecological inference based of gnawing damage (Haynes, 1980; Haynes, 1983). Indeed, bone-cracking is a specialized ecological strategy that has evolved several times within Carnivora (Van Valkenburgh, 2007; Tseng, 2013). While this gnawing behavior is most well-established in mammals —most commonly in orders Carnivora and Rodentia —this behavior may be present in other taxa as well.

In contrast to the pattern in Recent and Cenozoic mammals, most research on Mesozoic theropod dinosaurs has suggested that prey bone utilization in non-avian theropods is limited, with little direct evidence in the way of gnawing behavior (Fiorillo, 1991; Chure, Fiorillo & Jacobsen, 1998; Jacobsen, 1998; Hone & Rauhut, 2010). Patterns of tooth mark occurrence within Mesozoic assemblages support this interpretation in multiple ways. Firstly, relative to Recent and Cenozoic bone assemblages, there is a distinctly lower frequency of tooth marks in Mesozoic systems (Fiorillo, 1991; Hone & Rauhut, 2010). Additionally, these tooth-marked bone assemblages are dominated by scratch/scrape marks that do not penetrate the bone cortex, relative to puncture marks, suggesting these tooth-bone contacts are incidentally delivered while feeding on the surrounding soft tissue (Hone & Rauhut, 2010). Finally, documented instances of theropod tooth marks can generally be characterized by a single bite, inflicting either scratches or punctures to the bone, but not repeated bites in a restricted area (Chure, Fiorillo & Jacobsen, 1998; Hone & Rauhut, 2010). This suggests that prey bone utilization (i.e., gnawing, bone-breaking, osteophagy) is a key ecological strategy that was, and is, repeatedly exploited by mammalian carnivores, but not theropod dinosaurs (Hone & Rauhut, 2010).

A possible exception of this pattern is in the Tyrannosauridae where osteophagy may have been possible due to a combination of a strong bite and large, robust teeth (Hurum Jr & Currie, 2000; Hone & Rauhut, 2010; Gignac & Erickson, 2017). Despite this, direct evidence consistent with repeated biting on bones is rare (Erickson & Olson, 1996; Hone & Watabe, 2010; Dalman & Lucas, 2021), with the isolated, raking, and likely incidental marks dominating the tyrannosaur toothmark record. Putative tyrannosaur coprolites have demonstrated a high volume of consumed bone on occasion (Chin et al., 1998; Chin et al., 2003), though this may be more consistent with ingestion of intact portions of smaller prey animals.

Here we report an isolated hadrosaurid pedal ungual that shows strong evidence for repeated, powerful (i.e., sufficient enough to penetrate much of the bone cortex), and localized biting behavior in a small to medium-sized theropod dinosaur. The pattern of tooth marks is inconsistent with incidental contact, and rather, is a rare case of gnawing or ‘gnawing-like’ behavior in theropod dinosaurs.

Materials & Methods

The specimen, TMP 2018.012.0123 (Royal Tyrrell Museum of Palaeontology), is an isolated hadrosaurid pedal ungual, collected from Bonebed 50 (specifically, Bonebed 50 east) in the core area of Dinosaur Provincial Park, Alberta. The specimen was found as part of the Queen Mary University of London—Royal Tyrrell Museum of Palaeontology field school in 2018, and collected under Research and Collection Permit 18-510 (Alberta Tourism, Parks and Recreation) and permit to Excavate Palaeontological Resources 18-019 (Alberta Culture and Tourism).

Bonebed 50 is a mixed (macrofossil-microfossil) multitaxic bonebed in the lower to middle portion of the Dinosaur Park Formation, ∼19 m above the contact the underlying Oldman Formation. This site consists of a series of stacked paleochannel sandstones, the basal lags of each hosting a high diversity and abundance of microvertebrate fossil, as well as disarticulated to partially articulated adult hadrosaur skeletons, including the type of Corythosaurus excavatus (Tanke & Russell, 2012; Bramble et al., 2017), and isolated hatchling-to-nestling sized hadrosaur elements (Tanke & Brett-Surman, 2001; Eberth & Evans, 2011). Although at least three associated to articulated adult hadrosaur skeletons are known from this site, the ungual cannot be confidently associated with any of these, and likely represents an isolated specimen within the macrovertebrate component of the bonebed.

The ungual was almost completely encrusted with a soft to medium-hardness iron-rich siltstone. An air scribe on a low setting was gently used to remove most of this and a scalpel was used to remove the rest. The rock separated cleanly from the bone. An air abrasive on low air pressure and powder (sodium bicarbonate) flow settings was used as a final cleaning, followed by water and toothbrush. No adhesive, consolidant, or surface coat were applied.

Specimen photography was performed with a Canon EOS 6D (50 mm [1:1.4) and 24–105 mm [1:4] lenses). Ammonium chloride powder coating was used with photography to enhance the surface texture while homogenizing bone color. Ammonium chloride was applied using the “dry method” sensu Parsley, Lawson & PojetaJr (2018). All measurements were taken with digital calipers (150 cm) to the nearest tenth of a millimeter. Figures were prepared using Adobe Illustrator (V. 15.1.0) and Adobe Photoshop (V. 12.1).

Statistical test we performed in the R programing language (R Development Core Team, 2009) using the functions ks.test (stats), and chisq.test (stats), while the histograms were created using the function hist (graphics), lines (graphics) and density (stats). Chi-squared tests were used to compare the frequency of tooth marks between different skeletal elements (i.e., hadrosaur unguals to hadrosaur metapodials, hadrosaur unguals to all other hadrosaur elements). To see if the inter-tooth spacing of potential tracemakers were different from that of the the inter-mark spacing on the ungual, Kolmogorov–Smirnov tests (two sample) were used to test if these were drawn from a common size distribution.

Results

Description

The specimen is complete, missing only abraded portions of the cortical bone along its extremities, particularly on the rim of the articular facet (Fig. 1A). The ungual measures 105 mm in proximodistal length, 99 mm in maximum transverse width, and 49 mm it maximum height. The proximal articular face is 49 mm tall and 79 mm wide. The morphology and symmetry of the ungual indicates that it derives from digit three (i.e., III-4), the central and largest of the pedal digits (Zheng, Farke & Kim, 2011), and the largest ungual (Fig. 2A), although whether it is from the left or right side is unclear. Given that the specimen is equivalent in size, or larger than, specimens regarded as adults of contemporaneous hadrosaurid species (i.e., Gryposaurus, Corythosaurus), it likely pertained to an adult-sized individual (Parks, 1920).

Figure 1 Ammonium chloride powder coated photographs of the hadrosaurid pedal ungual TMP 2018.012.0123 showing bite marks (ventral/plantar view).

(A) View of entire specimen, with marks highlighted in blue (A’). (B) Close-up of the bitten region, with marks highlighted in blue and numbered in Arabic numerals (B’). All scale bars = 1 cm.

Figure 2 Position and orientation of bite marks on ungual and within hadrosaur pes.

(A) Right articulated hadrosaurid pes in dorsal view, with in ungual of digit three highlighted (white) and the position of the tooth marks (ventral side) indicated in black –modified from Prieto-Márquez (2014). (B) shaded line drawing of the ventral (plantar) view of the ungual TMP 2018.012.0123, showing the position of the bite marks (black). (C) Close-up view of bite mark size, shape, and orientation, showing alignment of bites in rows (Roman numerals) and columns (lowercase letters) indicated by dotted lines (based on Fig. 1B). (D) Close-up view of bite marks showing potential alignment of tooth row parallel with the long axes of the tooth marks. Hollow fills in C indicate potential bite marks missing from rows/columns. All scale bars = 1 cm.

A series of prominent tooth marks (observed in the field prior to collection) are present on the ventral (plantar) surface of the ungual, but no marks are seen of the dorsal or articular surface (Figs. 1A, 1B). A restricted area, ∼30 × 20 mm, on one-half of the ventral surface adjacent to the articular facet bears 13 distinct tooth marks (Fig. 2B). The largest tooth mark is 10.5 mm long, and 3.3 mm wide, while the smallest is 2.7 mm long and 1.6 mm wide (Table 1). The majority of the tooth marks are approximately three times longer than wide, but the smallest are more equidimensional. The morphology of the tooth marks is somewhat intermediate between the elongate v-cross section furrows, and the circular to ovoid pits that have previously been described for theropod tooth marks (Erickson & Olson, 1996; Jacobsen, 1998). Although prominent, the marks are shallow, with the deepest marks around 1 mm in depth. The marks penetrate the smooth surface of the cortical bone, and exposed the underling anteroposteriorly oriented cancellous bone fibers. Individual marks are numbered using Arabic numerals (Fig. 2C).

Table 1 Linear measurements of the 13 tooth marks on TMP 2018.012.0123. See Fig. 1B for mark numbers.

Mark	Row	Column	Length (mm)	Width (mm)	
1	i	b	7.8	2.7	
2	i	c	6.5	2.1	
3	ii	a	7.9	2.4	
4	ii	b	10.5	3.3	
5	ii	c	6.6	2.3	
6	iii	a	7.8	2.1	
7	iii	b	8.5	2.3	
8	iii	c	4.7	1.6	
9	iv	b	5.8	1.8	
10	iv	c	5.1	1.6	
11	v?	c?	2.7	1.6	
12	vi	a?	5.0	2.3	
13	vi	b?	4.0	2.3	
Mean	 	 	6.4	2.2	

Relative to each other, the marks are not random in orientation or position. The long axis of all tooth marks is parallel, running ∼20° to the transverse axis of the digit (Figs. 1A, 2B). Further, the marks are positioned in an approximate grid pattern, being aligned in two nearly perpendicular axes (Fig. 2C). The long axes of the marks are nearly aligned with one of these grid axes (oriented distomedially) termed rows (labeled with Roman numerals), and lie nearly perpendicular to the other (oriented distolaterally) termed columns (labeled with lowercase letters) (Fig. 2C). All tooth marks, with the exception of mark number 11, fit this discrete grid-like pattern. There are three columns (a-c) and at least four, but possibly up to six rows (i-vi). Tooth mark spacing (based on midpoints) between successive marks within rows varies from 4.9 to 8.8 mm, with a mean of 7.0 mm, while spacing between successive marks in columns is smaller, from 4.0 to 7.4 mm with a mean of 5.3 mm (Table 2). The bone surface on which the marks occur is slightly convex along a transverse transect. In contrast an orthogonal proximodistal transect the surface show more topographic variation, and specifically is strongly concave at the distolateral extreme of the element.

Table 2 Spacing between successive tooth marks by both rows (i, ii, etc.) and columns (a, b, etc.) on TMP 2018.012.0123, see Fig. 2C.

Alignment	Marks	Distance (mm)	
Rows	 	 	
i	1, 2	8.8	
ii	3, 4	8.6	
ii	4, 5	6.8	
iii	6, 7	6.6	
iii	7, 8	4.9	
iv	9, 10	6.4	
	Mean	7.0	
Columns	 	 	
a	3, 6	6.1	
b	1, 4	7.4	
b	4, 7	4.9	
b	7, 9	4.2	
c	2, 5	4.0	
c	5, 8	5.5	
c	8, 10	5.2	
	Mean	5.3	

Frequency of tooth marks on unguals

Despite the apparent oddity of bite marks to a hadrosaur ungual, TMP 2018.012.0123 is not an isolated occurrence. A second hadrosaur pedal ungual from the Dinosaur Park Formation, UALVP 55092 (University of Alberta Laboratory of Vertebrate Paleontology), appears to shows a cluster of three distinct tooth marks on the ventral (plantar) portion of the ungual, oriented at ∼45° to the long axis of the ungual (Fig. 3A). In this second case, the marks appear to be a series of three parallel shallow furrows consistent with a single bite, and more in line with other described theropod feeding traces.

Figure 3 Photographs (upper) and interpretive drawings (lower) of three isolated hadrosaurid pedal unguals with theropod tooth marks.

(A) UALVP55092 in ventral (plantar) view, (B) TMP 1979.008.0769 in dorsal view, (C) TMP 1980.016.1215 in dorsal view. In lower drawings hatched areas indicate missing bone surface, dashed lines are approximate margins, and black indicates tooth marks. All specimens to same scale. Scale bar = 1 cm. Photograph in Fig. 3A courtesy of Ian Macdonald.

Given that two Dinosaur Park Formation hadrosaur unguals show unexpected bitemarks, a survey was undertaken to determine if this is a more common, but previously unrecognized, pattern. A total of 425 isolated hadrosaurid unguals (pedal and manual) from the Dinosaur Park and Oldman formations across southern Alberta, were specifically examined for tooth marks (Table S1). Only two cases of definitive tooth marks were found within this sample (Figs. 3B, 3C), suggesting the frequency of tooth marks on hadrosaur unguals is very low (<1%). This is significantly lower than the reported frequency of tooth marks on both overall hadrosaurid bones, 14% (47/339), and hadrosaur metapodials, 13% (16/120), from this formation (chi-square = 54.152 and 44.535, p = 1.856e−13 and 2.499e−11, respectively) (Jacobsen, 1998).

These two other tooth marked unguals are smaller than TMP 2018.012.0123, and, as with UALVP 55092, the marks are clusters of parallel (or nearly parallel) elongate furrows oriented at ∼45° to the long axis of the ungual. The first of these specimens, TMP 1979.008.0769 (II-3?), is 59 mm wide and 74 mm long, and preserves four tooth marks (11.0, 23.4, 25.1, and 10.9 mm in length) on the dorsolateral surface with inter-mark spacing of around 3.5 mm (Fig. 3B). The second specimen, TMP 1980.016.1215 (III-4), is 59 mm wide and 64 mm long (estimate), and also preserves four tooth marks (9.4, 7.5, 26.7, 8.1 mm in length) on the dorsal surface with inter-mark spacing of around 5.2 mm (Fig. 3C). In all four cases, marks are seen one on side of the ungual and not the other.

The frequency of tooth marks in non-ungual phalanges may be higher than for unguals, as several cases are known and have been reported for both hadrosaurid—TMP 1966.011.0022 (Jacobsen, 1998), TMP 1981.023.0011 (Jacobsen, 1998), TMP 1993.110.0003, TMP 2016.012.0096, TMP 2019.014.0004, UCMP 140601, (Erickson & Olson, 1996)—and tyrannosaurid—UCMP 137538, MOR 1126 elements (Longrich et al., 2010). An allosaur pedal ungual with allosaur tooth marks has also been recently reported (Drumheller et al., 2020).

Discussion

Pattern of tooth marks

In total, 13 distinct tooth marks are preserved on TMP 2018.012.0123, but their alignment in a grid-like pattern suggests these were the result of a combination of successive teeth in the toothrow making contact with the bone surface, and the tooth row moving laterally relative to the ungual between successive bites.

Given the alignment of the mark long axes, the relatively consistent within-row mark spacing, and the saddle shaped bone surface, it is likely that the rows (e.g., i, ii, iii etc.) represent individual bites, with successive teeth in the tooth row making aligned marks (Fig. 2C). The multiple rows represent the bone being moved laterally relative to the tooth row (or vice versa) between successive bites. The individual marks within columns (e.g., a, b, c) therefore are interpreted as the same tooth making contact with the bone surface multiple times as the bone slid laterally relative to the tooth row between bites. Under this hypothesis there are up to three successive teeth in the tooth row that make contact with the bone, and a minimum of four, and possibly up to six, distinct bites. The relatively equal spacing between the successive bites (i.e., rows i–iv) indicate that the relative change in the position of the bone to the teeth differed by consistent distance between each successive bite, ∼5 mm. Under this hypothesis, the average distance between successive teeth in the toothrow is 7.0 mm. A slightly different hypothesis is illustrated if the long axis of the marks, rather than the position of the marks, are used as the primary alignments (Fig. 2D). Under this hypothesis lines illustrating possible tooth marks aligned in the toothrow are indicated. For this hypothesis, marks caused by successive teeth, and therefore the spacing between them, are less obvious to establish, but are roughly similar to that of the scenario above.

It is possible, though in our mind less likely, that the interpretation of these axes may be swapped. In this case, the columns (e.g., a, b, etc.) represent successive teeth in the toothrow making contact in a single bite, while the rows (e.g., i, ii, etc.) represent repositioning and lateral movement of the bone between successive bites. Under this hypothesis there are up to five successive teeth (with a potential one-tooth gap) in the tooth row that make contact with the bone, and only three distinct bites. Under this hypothesis, the average distance between successive teeth in the toothrow is much smaller at 5.3 mm. Movement of the teeth across the bone surface during the bite in this hypothesis would be nearly orthogonal to the cross-sectional long axis of the tooth.

Several lines of evidence support the former interpretation relative to the later. The relative size and shape of the marks is more consistent within columns (e.g., marks 1, 4, 7) than within rows (e.g., marks 3, 4, 5). The spacing between successive marks is more consistent within rows, than within columns. The movement orthogonal to the cross-sectional long axis of the tooth would also increase the chance of tooth damage (see Hone & Chure, 2018). Finally, the bone surface transected by the rows (i.e., i, ii) is gently convex, with the bites located at the high point of the transect. Conversely the bone surface transected by the columns (i.e., a, b) has a higher amplitude topography, is broadly concave, and bounded proximally and distally by bone surfaces that are above the level of the tooth marks, but that are not marked.

Regardless of which of these biting scenarios is correct, a series of closely spaced powerful bites were delivered to the ungual with the element repositioned relative to the tooth row between successive bites.

Tracemaker

The tracemaker responsible for the toothmarks can be narrowed down to a relatively small number of candidates. The various non-dinosaurian carnivores present in the Dinosaur Park Formation assemblages, including mammals, crocodylians and squamates, can be ruled out—see similar discussion in Hone, Tanke & Brown (2018). Gnawing marks thought to derive from mammalian tracemakers have been described from the Belly River Group, and these broadly resemble gnaw marks of modern rodents (Longrich & Ryan, 2010). The bite marks of both modern and Cretaceous crocodylians leave characteristic deep, circular to sub-circular punctures (Njau & Blumenschine, 2006; Noto, Main & Drumheller, 2012; Boyd, Drumheller & Gates, 2013; Botfalvai, Prondvai & Osi, 2014; Drumheller & Brochu, 2014) distinct from those seen on TMP 2018.012.0123. Although furrows with V-shaped cross-sections can be produced by crocodylian teeth (Njau & Blumenschine, 2006), these are neither characteristic nor diagnostic of this group. They are also often continuous with bisected pits, represent significant movement of the tooth across the bone surface, and there is no indication of systematic biting, making a crocodylian a poor match to the regular marks on TMP 2018.012.0123. Finally, the tooth marks left by modern large squamates are dominated by thin arcing scours, with rare pits and no crushing observed (D’Amore & Blumenschine, 2009). None of these features are consistent with the marks left on TMP 2018.012.0123.

Within Dinosauria, several clades of carnivorous (and potentially omnivorous) theropods represent potential tracemakers, including Tyrannosauridae, Dromaeosauridae, and Troodontidae. Of these potential tracemakers, Tyrannosauridae has the most comparative material for bite traces both in terms of described material and absolute number of marks. Tooth marks thought to have been delivered by tyrannosaurs are dominated by v-shaped furrows and scours (both (sub)parallel and isolated), as well as distinct puncture-and-drag marks, punctures, and fine parallel striae resulting from denticle scrapes (Jacobsen, 1995; Erickson & Olson, 1996; Chin, Farlow & Brett-Surman, 1997; Jacobsen, 1998; Fowler & Sullivan, 2006; Hone & Watabe, 2010; Bell, Currie & Lee, 2012; Rivera-Sylva, Hone & Dodson, 2012; Robinson, Jasinski & Sullican, 2016).

Dromaeosaurid tooth marks reported include those of Saurornitholestes, on the tibia a large azdarchid pterosaur (Currie & Jacobsen, 1995), those of a velociraptorine on Protoceratops bones, (Hone et al., 2010), and those thought to pertain to Deinonychus antirrhopus on the skeleton of Tenontosaurus (Gignac et al., 2010). In the former two cases the marks are shallow grooves or scours, while the latter case these are deeper V-shaped furrows. In addition to the scours and furrows, deeper, semi-circular ‘bite and drag’ marks have also been observed in velociraptorine (Hone et al., 2010), while deep punctures are described for Deinonychus (Gignac et al., 2010).

Fewer tooth marks attributable to Troodontidae are known, but Jacobsen & Bromley (2009) describe an example of the trace Linichnus serratus which they attribute to Troodon sp. Given their close relationship and general similarity to dromaeosaurs, they might, however, have left similar traces if they bit into bones.

Given the lack of evidence of denticle spacing present on bite marks, and that both Tyrannosauridae and Dromaeosauridae were capable of delivering bites resulting in deep furrows and pits to the bone surface, the relative size and shape of the tooth marks, and the spacing between these marks may help do determine which is a more likely tracemaker. As noted by Hone & Chure (2018), drawing direct correlation between spacing of (presumed) serial tooth marks and tooth spacing in potential tracemarkers may be problematic. Factors such as curved bone surfaces, bite angle, and missing or misaligned teeth may add additional variation to the resultant tooth mark spacing and may make elimination of potential tracemarkers more challenging (Hone & Chure, 2018). However, when a relatively consistent pattern of spacing between aligned tooth marks is observed, a most parsimonious first assumption is that this spacing is at least coarsely comparable to the spacing of teeth in the trace-making individual (Jacobsen, 2003).

Table 3 Spacing between successive tooth positions (or alveoli) in tooth rows across specimens of several potential tracemakers.

Taxon	Specimen	Element																	Mean (mm)	Count (n)	
Gorgosaurus libratus - juv.	TMP 1994.012.0155	R. Dent.	12.6	12.3	12.9	12.8	11.9	12.8	12.1	12.4	12.7	10.3	9.6						12.0	11	
Gorgosaurus libratus - juv.	TMP 1994.012.0155	L. Dent.	11.4	12.5	14.2	12	12.9	12	12.1	11.2	10.7	10.1							11.9	10	
Gorgosaurus libratus - juv.	TMP 1990.081.0006	R. Dent.	10.7	10.5	11	11.6	11	10											10.8	6	
Saurornitholestes langstoni	TMP 1988.121.0039	L. Dent.	5.8	4.9	5	5.6	6.4	5.1	5.2	4.9	5	5.9	5.9	4.9	4.3	4.4			5.2	14	
Saurornitholestes langstoni	TMP 1991.036.0112	L. Dent.	7.2	7.6	7.9	7.8													7.6	4	
Stenonychosaurus inequalis	TMP 1967.014.0039	L. Dent.	2.2	2.2	2.1	2.6	2.7	2.5	2.6	2.5	2.6	2.3	2.8	2.7	2.3	2.9	2.9	3.1	2.6	16	
Stenonychosaurus inequalis	TMP 1982.016.0138	L. Dent.	3.6	3.6	3.6	3.1	3.4	3.1											3.4	6	
Dromaeosaurus albertensis	TMP 1984.008.0001	L. Dent.	6.3	8.3	8.2	8.1	7.9	7.8	7.3	7.6	8.5								7.8	9	
Dromaeosaurus albertensis	TMP 1984.008.0001	R. Dent.	6.7	6.6	8.3	8.5	8.7	8.6	8.2	7.6	7.4	6.6							7.7	10	

Table 4 Results of Kolmogorov–Smirnov test (two sample), comparing the spacing between successive tooth positions (or alveoli) in TMP 2018.012.0123 with specimens of several potential tracemakers.

			Toothmark Spacing	
Potential tracemakers tooth spacing	Rows	Columns	
Specimen	Taxon	N	6	7	
TMP 1967.014.0039	Stenonychosaurus inequalis	21	0.000324	0.000118	
TMP 1982.016.0138	Stenonychosaurus inequalis	6	0.00496	0.00313	
TMP 1988.121.0039	Saurornitholestes langstoni	14	0.0153	0.841	
TMP 1991.036.0112	Saurornitholestes langstoni	4	0.181	0.0303	
TMP 1984.008.0001	Dromaeosaurus albertensis	16	0.299	0.00109	
TMP 1990.081.0006	Gorgosaurus libratus - Juvenile	6	0.00496	0.00313	
TMP 1994.012.0155	Gorgosaurus libratus - Juvenile	19	0.000177	5.51E−05	

Figure 4 Size comparison of spacing between subsequent tooth marks on TMP 2018.012.0123 (A, B, J), and exemplars of potential theropod trace making taxa (C–I, K–N).

(A–I) Histograms showing distributions on spacing between tooth marks (A, B) and teeth/alveoli (C–1): (A, B) TMP2018.012.0123, for rows (A) and column (B); (C, D) Stenonychosaurus inequalis – TMP 1967.014.0039, and 1982.016.0138; (E, F) Saurornitholestes langstoni—TMP 1988.121.0039 and 1991.036.0112; (G) Dromaeosaurus albertensis—TMP 1984.008.0001 (cast of AMNH 5356), and H, I) juvenile Gorgosaurus libratus—TMP 1990.081.0026 and 1994.012.0155. J–N) Scaled line drawings illustrating the morphology and size of the tooth traces exemplar dentaries (J) ungual TMP 2018.012.0123 in ventral view, with tooth marks shown in black; (K) medial view of reconstructed Stenonychosaurus inequalis (Troodontidae) dentary based on CMN 8540, redrawn from (Currie, 1987); (L) medial view of complete dentary of Saurornitholestes langstoni (Dromaeosauridae) –based on TMP 1988.121.0039; (M) lateral view of complete dentary of Dromaeosaurus albertensis (Dromaeosauridae) –based on AMNH 5356, redrawn from (Currie, 1995), (N) medial view of a dentary of a juvenile Gorgosaurus libratus (Tyrannosauridae) - based on TMP 1994.012.0155. Lines above dentaries indicates tooth size/alveolar spacing. Significance values: * α = 0.05, ** α = 0.01, *** α = 0.001, ns = not significant –See Table 4. All specimens to same scale. Scale bar = 1 cm.

Comparisons between the spacing between the tooth marks on the ungual (Table 2), and the inter-tooth spacing of potential theropod tracemakers (Table 3) is shown in Fig. 3. The teeth (or alveoli) of the exemplar dentaries of the troodonid Stenonychosaurus inequalis average 2.7 or 3.4 mm in inter-tooth spacing (Table 3), significantly more closely spaced that the tooth marks in TMP 2018.012.0123 (Table 4, Figs. 4C, 4D).

For the two dromaeosaurid taxa, both Saurornitholestes langstoni and Dromaeosaurus albertensis are known from specimens that overlap the size range of the tooth marks on TMP 2018.012.0123. The Saurornitholestes dentary TMP 1988.121.0039 (mean inter-tooth spacing = 5.2 mm) is not statistically different from the within-column tooth mark spacing (Table 4, Fig. 4E), while a larger specimen, TMP 1991.036.0112, (mean inter-tooth spacing = 7.6 mm) is not statistically different from the within-row tooth mark spacing (Table 4, Fig. 3F). Similarly, the best specimen of Dromaeosaurus, AMNH 5356 (cast, TMP 1984.008.0001) shows inter-tooth spacing (mean = 7.7 mm) that is not statistically different from the within-row tooth mark spacing (Table 4, Fig. 4G).

Relative to both Troodontidae and Dromaeosauridae, Tyrannosauridae is much better sampled from the Belly River Group, with a nearly complete ontogenetic series of jaws, missing only the smallest size classes. Given the small size of the tooth marks, and their close spacing only small, immature, individuals could represent potential tracemarkers for tyrannosauridae. The two smallest tyrannosaurid jaws from the Dinosaur Park Formation, the dentaries TMP 1990.081.0026 (mean spacing = 10.8) and TMP 1994.012.0155 (mean spacing = 12.0 mm), both show tooth spacing that is significantly greater than the spacing observed in TMP 2018.012.0123 (Table 4, Figs. 4H, 4I). While both these specimens are young juveniles (tooth row length of 2018.012.0123 = 165 mm) they do not represent the youngest/smallest extreme of tyrannosaurid ontogeny. It is possible that a younger/smaller tyrannosaur may have made the marks on TMP 2018.012.0123, but this would have to be a very immature individual smaller than that of any jaw currently know from the Belly River and Edmonton groups –with a tooth row length <165 mm.

Given these data, it is not possible to confidently determine the taxonomy (or ontogeny) of the tracemarker, but it can likely be narrowed down to either and adult-sized dromaeosaurid, or a very young tyrannosaurid. Regardless, these tooth marks suggest a potentially novel bone utilization, and may expand the evidence for bone utilization in Tyrannosauridae and/or Dromaeosauridae.

Behavioral hypotheses

Several possible behavior explanations may be put forward to explain the occurrence of such distinct tooth marks on the ungual.

Incidental contact while feeding

Regardless of whether the ungual was articulated with, or isolated from, the body, it would have had little meat in close association and represent a ‘low economy’ element (sensu Drumheller et al., 2020). While hadrosaur footprints indicate fleshy pads under the pedal phalanges (LangstonJr, 1960; Currie, Nadon & Lockley, 1991), these would not have extended to the ungual, which would have largely been covered in a kertatinous hoof. Although a keratinous hoof on the ungual would have had a very high protein content, keratin is resistant to vertebrate digestion, and was likely not a high value food item (Bragulla & Homberger, 2009). Indeed, there are few bones in a hadrosaur skeleton that would be either less desirable for consumption, or further from areas of high consumption priority. Actualistic taphonomic studies of carcass utilization by modern mammalian carnivores consistency recover the phalanges and unguals as being bones with some of the lowest frequency of modification (Domınguez-Rodrigo, 1999; Rodríguez-Hidalgo et al., 2013; Arilla et al., 2014; Rodríguez-Hidalgo et al., 2015; Arilla, Rosell & Blasco, 2019) and these elements rank low in the carcass consumption sequence, and are largely used for their marrow (Blumenschine, 1986; Marean et al., 1992). Additionally, the tooth marks on the bone surface of TMP 2018.012.0123 are not consistent with glancing contact between tooth and bone, but appear to be as a result of directly biting the bone surface. These tooth marks are not consistent with incidental marks during feeding, which appears to be the case for the majority of theropod tooth marks (Hone & Rauhut, 2010).

Predation/Grasping

It is possible that the bites were delivered to the prey animal while it was still alive, and are the result of active predation. In this hypothesis, the predator may have latched onto the hind foot of the hadrosaur with its jaws in an effort to it slow down and, presumably with the combined effort of multiple individuals, bring down the prey. Multiple bites, and repositioning of the tooth row between bites, may be indicative of the struggle between prey-and-predator.

There are several problems with this hypothesis. Firstly, the differential between potential predator and prey size is extreme. Mass estimates for adult-sized potential tracemakers range from 16 kg and 18 kg for Dromaeosaurus albertensis and Saurornitholestes langstoni, to 57 kg for Troodon inequalis (Campione et al., 2014; Benson et al., 2018). Mass estimates for an immature tyrannosaur are more challenging. Given scaling of the skull length to body mass (Therrien & Henderson, 2007) in Theropoda, and femur to skull length in Tyrannosauridae (Currie, 2003), and mass and femur length (Christiansen & Farina, 2004), the lower jaw of TMP 1994.012.0155 (29 cm long) would suggest a tyrannosaur tracemaker was no more than 32 kg (Therrien & Henderson, 2007) or 44 kg (Christiansen & Farina, 2004). It should be noted that these estimation methods are not designed for immature individuals, are likely underestimates, and should be regarded as coarse at best. In comparison, adult-sized hadrosaurid taxa from Dinosaur Park Formation have mass estimates ranging from >3,000 to >5,000 kg (Campione & Evans, 2012; Benson et al., 2018). This puts the mass of the hadrosaur at two orders of magnitude greater than the putative theropod tracemakers. This size differential is much greater than that seen between predator and prey in analogous systems where typically the predator is much larger than the prey (Hone & Rauhut, 2010). Given this size differential, it is difficult to believe that a theropod grabbing the rear ungual of a hadrosaur could not easily be kicked off. Further, if the multiple marks are interpreted as the result of a moving/struggling prey animal, one would expect there to be slippage and rotation of the marks, and the spacing and alignment of successive to be more irregular. Rather, the multiple bites are parallel and equidistant.

Play

Tyrannosaur tooth marks on isolated bones have been interpreted as evidence for play (Rothschild, 2015). This hypothesis has been reviewed (Snively & Samman, 2015), and it has been pointed out that it makes few testable predictions, and is difficult to falsify. Object based play behavior is within the behavioral extant phylogenetic bracket for Dinosauria (Snively & Samman, 2015), so this behavior in a theropod dinosaur may not be unexpected. Given its difficulty to test, however, this hypothesis is not addressed in detail here, though we suggest that the repeated nature of the bites at a single location on the ungual do not easily align with the idea of play.

Late-stage carcass consumption

Perhaps the most likely hypothesis is that the tooth marks are result of late-stage carcass consumption (Hone & Watabe, 2010) (Fig. 5). Repeated, high powered bites delivered near the articular face of the ungual may have served to either sever or disarticulate the bone from the rest of the foot, or to break open the bone as part of a bone consumption strategy. The multiple parallel marks may be consistent with repositioning of the bite to produce better leverage as the tracemaker attempted to pull apart the skeleton. The hypothesis of late-stage carcass consumption is consistent with interpretations of other densely tooth marked specimens attributable to tyrannosaurs (Erickson & Olson, 1996; Fowler & Sullivan, 2006; Hone & Watabe, 2010).

Figure 5 Artistic reconstruction of one potential scenario causing the tooth marks on TMP 2018.012.0123.

A juvenile tyrannosaurid bites down on a hadrosaur ungual as part of a late-stage carcass consumption strategy. Artwork courtesy of Joshua Doyon.

This specimen, however, differs from these other reports in several major ways. First, the ungual is likely to have little to no flesh in association, especially compared to a ceratopsian pelvis (Erickson & Olson, 1996; Fowler & Sullivan, 2006) or hadrosaur humerus (Hone & Watabe, 2010). As such, this may represent an even more extreme example of late-stage carcass consumption than these previous reports. Second, although these other specimens show a high number of tooth marks, and are consistent with multiple bites to a single bone, these bites (with the possible exception of the deltopectoral crest of the humerus Hone & Watabe, 2010) are not delivered to the same areas repeatedly. The marks to TMP 2018.012.0123 are restricted to a small area that was bitten up to six times. Third, previous records of bones with a high density of tyrannosaurid bite-marks are attributable Tyrannosaurinae; i.e., Daspletosaurus (Fowler & Sullivan, 2006), Tarbosaurus (Hone & Watabe, 2010), Tyrannosaurus (Erickson & Olson, 1996). The tooth mark described here, may be attributable to either Albertosaurinae (i.e., Gorgosaurus) or Tyrannosaurinae (i.e., Daspletosaurus) or to Dromaeosauridae, potentially broadening this behavior phylogenetically within Theropoda. Finally, these other reports document feeding behavior in adult-size tyrannosaurs. If the bite marks described herein are from a tyrannosaur, they are from a very small, young individual, an individual at, or below, the size of the smallest known articulated skulls. It is unclear, however, if the purpose of the repeated, localized bites was to dismember the ungual from the rest of an articulated foot to expose articular cartilage or tendon, to break open the bone to exposed the marrow, or for some other purpose.

Bone utilization by theropod dinosaurs

A recent study on theropod tooth mark frequency on bones from the Upper Jurassic Mygatt-Moore Quarry (Drumheller et al., 2020), showed a high mark frequency on similar ‘low economy’ bones (e.g., equivalent to the ungual), a frequency that was higher than ‘high economy’ bones. This suggested high rates of scavenging and potentially representative of a highly-stressed environment. While this is consistent with the marks seen in TMP 2018.012.0123 this may be an exception as the frequency of marks on other unguals and ‘low economy’ bones appears to be quite low, and in this regard the pattern of bite marks in the Dinosaur Park Formation is largely distinct from the Mygatt-Moore Quarry (Jacobsen, 1998).

We suggest that the general lower levels of bone exploitation by non-avian theropods may be linked to their difficulty of accessing the marrow cavity. Mammals may have proportionally larger marrow cavities than do dinosaurs (particularly ornithischians) for a given bone diameter, but in any case, large dinosaurs will have absolutely thicker bone walls compared to mammals (e.g., sauropods and large ornithischians have absolutely larger femora than any living terrestrial mammals aside from, perhaps, elephants). Furthermore, large mammalian carnivores typically have more robust teeth for their size than do non-tyrannosaurid theropods so overall would have a greater ability to process bone to obtain the marrow than most theropods. Together these factors may explain the distinction in bone utilization between the two clades. The bones of dinosaurian juveniles or small taxa could still be broken and /or consumed and thus destroyed, but the lack of bite traces on large dinosaurian elements may at least in part reflect an inability to break into them.

It should be noted that some living birds (extant avian theropods), while not engaging in gnawing behavior on bones, do make extensive use of bone as a dietary source. The most extreme example of this is the bearded vulture (Gypaetus barbatus), for which bones make up the majority of the diet, and generally derive from much larger animals (Brown & Plug, 1990; Margalida, Bertran & Heredia, 2009). As such, bone utilization strategies across Theropoda be more complex then previously realized.

Conclusions

A hadrosaurid pedal ungual bears a distinct pattern of tooth marks suggesting multiple, repeated, bites delivered to a restricted area of the element, and powerful enough the penetrate much of the bone cortex. The morphology, size and spacing of the tooth marks suggest the tracemaker was a small-to-medium sized theropod dinosaur, likely a dromaeosaurid or young tyrannosaur. This behavior is most consistent with late-stage carcass consumption of an element that had limited association with the soft tissue considered to be the primary food source. Theropod bite marks on other hadrosaur unguals are known, but appear to occur at a very low frequency.

The traces left on these unguals, particularly TMP 2018.012.0123, are largely consistent with those left by gnawing behavior. The mechanism of the bone processing behavior is at least superficially similar to gnawing in mammals, and may represent “gnawing-like” behavior. Indeed, if similar marks were left on a bone from a mammalian carnivore, “gnawing” would likely be considered an appropriate term. This specimen expands our understanding of prey bone utilization behavior in non-avian theropods dinosaurs, representing the strongest case for “gnawing-like” behavior in this clade. The occurrence of this prey bone utilization is also expanded, either phylogenetically into Dromaeosauridae, or ontogenetically into young tyrannosaurids.

Supplemental Information

Supplemental Information 1 Survey of isolated Dinosaur Park and Oldman formation hadrosaur unguals for tooth marks

TMP - Royal Tyrrell Museum of Palaeontology, UALVP - University of Alberta Laboratory of Vertebrate Paleontology.

Click here for additional data file.

TMP 2018.012.0123 was found by EL Clare, as part of the Queen Mary University of London—Royal Tyrrell Museum of Palaeontology field school in 2018. The authors thank Alberta Parks staff, specifically J Blacklaws, for logistical support of this field school. UALVP 55092 was collected by C Coy, and TMP 1979.008.0769 and TMP 1980.016.1215 were collected by J Bahr and P Currie respectively. Access to collections and specimen assistance was facilitated by T Courtenay, H Feeney, R Russell, B Sanchez and B Strilisky (RTMP), and H Gibbins, C Coy and P Currie (UALVP). D Brinkman provided materials and assistance for ammonium chloride powder coating. I Macdonald brought UALVP 55092 specimen to our attention, and provided the photograph. D Evans, N Gardner and F Varriale provided useful discussion. J Doyon created the artwork in Fig. 5. P Bell, J Farlow, D D’Amore (reviewers) and F Knoll (academic editor) provided feedback which improved the paper.

Additional Information and Declarations

Competing Interests

Author Contributions

Field Study Permissions

Data Availability

The authors declare there are no competing interests.

Caleb M. Brown conceived and designed the experiments, performed the experiments, analyzed the data, prepared figures and/or tables, authored or reviewed drafts of the paper, and approved the final draft.

Darren H. Tanke conceived and designed the experiments, authored or reviewed drafts of the paper, and approved the final draft.

David W.E. Hone conceived and designed the experiments, analyzed the data, authored or reviewed drafts of the paper, and approved the final draft.

The following information was supplied relating to field study approvals (i.e., approving body and any reference numbers):

TMP 2018.012.0123 was collected from Dinosaur Provincial Park, Alberta, under Research and Collection Permit 18-510 (Alberta Tourism, Parks and Recreation) and permit to Excavate Palaeontological Resources 18-019 (Alberta Culture and Tourism), both issued to Caleb M. Brown.

The following information was supplied regarding data availability:

Raw data for measurements of tooth marks are available in Tables 1–2. Raw data for measurements of tooth/alveolar spacing of potential tracemakers are available in Table 3. Raw data for tooth mark frequency in isolated hadrosaur unguals are available in Table S1.

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
