# Peer review of "Rare evidence for ‘gnawing-like’ behavior in a small-bodied theropod dinosaur"

_PeerJ, doi:10.7717/peerj.11557_

## Round 0.1 · original submission · Minor Revisions

The reviewers are positive about your work, but still offer some suggestions for improvements. Careful editing is needed.
Please, together with your unmarked revised manuscript, provide a marked-up copy as well as a document explaining how you have addressed each of the points raised by the reviewers.

·

Basic reporting

Generally clear and professionally written (see general comments below)

Experimental design

no comment

Validity of the findings

no comment

Additional comments

Review of Brown et al

Brown et al. present a logical interpretation of bite marks on a hadrosaur ungual as repetitive ‘gnawing-like’ behaviour by a non-avian theropod. This potential behaviour has not been observed before in a non-avian theropod and therefore provides a novel insight into their feeding strategies. In all, I find the identifications and justifications for the various interpretations are reasonable. There are, however, a large number of typos etc that I have noted on the marked-up doc (attached) as well as a number of more detailed comments/queries that need to be addressed. Several additional comments are provided here:

Lines 17, 36 etc: A constant struggle with interpreting bitemarks in the fossil record is whether they are predated upon or simply scavenged. In general, this requires a bit more caution in the wording throughout the manuscript, saying something like “feeding strategy” rather than “predation” (I acknowledge there are some places where ‘predation’ is also used correctly, eg. Line 313). It also seems at odds with the overall message of the manuscript, which is that some theropods employed gnawing-like behaviour, presumably for mineral supplements. While predatory mammals also gnaw, it is also widespread in non-predatory mammals (rodents etc). While I have no doubt dromaeosaurs were predators, you’re unlikely to be able to make this distinction from a single ungual. However, given that the ungual is apparently from a full grown (or large-bodied) individual, predation is arguably less likely as you point out in the Discussion. So, if you do want to stick your neck out later on in the manuscript and “say” something about the feeding strategy, then scavenging might be the more appropriate option. But in the intro, I'd stick with something more generic like “feeding behaviour/strategy”.
This issue continues to mammals (modern and fossil), as gnawing behaviour is associated with mineral supplements in a wide range of non-carnivorous mammals. Therefore, ‘predations’ in the context of mammals is also incorrect.

Lines 35-36. Even though I think we can agree that birds don’t gnaw, certain vultures (e.g. bearded vulture) will swallow whole bones which certainly counts a prey bone utilisation. Please be clear you are talking about non-avian theropods, not just ‘theropod dinosaurs’. This distinction should also be made in the abstract and elsewhere.

lines 225-228: Njau and Blumenschine (2006) also report V-shaped clefts as distinctive crocodylian feeding traces. The cross sectional morphology of the marks on the ungual were not described, so it is difficult to judge, but marks #3 and maybe #6 look like they might have V-shaped cross sections based on the figure. Some attention to the ungual description and comparison to this type of croc trace (in the Discussion) is required to categorically rule out crocodylian feeding traces.

L248-250: Absence of Troodon feeding traces. Jacobsen and Bromley (2009) identified bite marks that they attribute to Troodon on bone from DPP. I think he also described additional material in his thesis but I wasn’t able to double check that…
Jacobsen, A. R., & Bromley, R. G. (2009). New ichnotaxa based on tooth impressions on dinosaur and whale bones. Geological Quarterly, 53(4), 373-382.

L 375-385: This final paragraph seems like a rather disjointed series of thoughts rather than a coherent summary or “future directions” statement. It might be better to phrase it as a series of hypotheses that might be tested with the proverbial “new discoveries”.

·

Basic reporting

There were a lot of typographic errors, misspellings, and grammatical errors in the manuscript as received. I have correctly all of the mistakes that I caught.

I have scribbled several queries about wording in the attached PDF.

Otherwise no major problems.

Experimental design

This all seems pretty reasonable.

Validity of the findings

Minor comments and queries are indicated on the attached, edited (by me) manuscript.

The findings are well-presented, and seem reasonable to me.

The only substantive query I have is about the authors' contention that the bites involved in marking the hadrosaur ungual were forceful. If the bite marks are only 1 mm deep (is that another typographical error? Did the authors mean 1 cm deep?), that doesn't seem to me like it would involve a lot of force.

Additional comments

I have nothing further to add, apart from my editorial markings and questions on the ms itself.

·

Basic reporting

The manuscript is clear and well written. There are a fair amount of typos, but nothing that cannot be easily fixed. These did not impede my ability to understand what was trying to be said. The background is thorough, although I would have liked a few more references concerning mammalian gnawing, as it is the standard by which the conclusions are based. The structure of the article is professional and well put together, and is appropriate in length. Figures, tables, and raw data are professionally presented.

Experimental design

The study has a clear purpose and was conducted thoroughly. It elaborates upon new evidence which suggests novel behavior in theropod dinosaurs. All permits appear to be in order. Although the methods appear to be sound, some of them are not mentioned in the actual Materials and Methods section. These should be included upon revision and explained thoroughly.

Validity of the findings

The authors make a clear and very compelling argument, which is an important addition to the state of the science. There conclusions are appropriate and speculations do not overreach. The authors account for most alternative explanations for the data presented, although a stressed environment may provide further explanation for the consumption of bones in low economy body regions.

Additional comments

This is a well put together report on an unique bone trace that suggests novel behavior in theropods reminiscent of mammal carnivores. It is an important addition to the state of the science concerning Mesozoic taphonomy. No major changes are necessary, but there are a number of minor changes that are needed before publication.

All the methods seem appropriate to the study, but many are not actually mentioned in the Materials and Methods section. The programs used for the statistics are mentioned (line 104), but the tests themselves are not mentioned (such as the chi^2 and K-S tests), nor is a rational for why they are being used or what they are testing. The correlation between mark spacing and the teeth of fossil taxa is not mentioned either, and, even though it is explained in the legend of figure 4, it should be outlined in the M&M section instead/as well.

A few more references concerning mammalian feeding traces and carcass consumption sequences may be added if desired. Please see the works of Binford, Hill, Capaldo, Marean, and/or Pobiner.

Drumheller et al., 2020 also addresses marks occurring on low economy bones such as terminal pedal digits. Could gnawing on ornithopod unguals also indicate a high stress environment as they postulate for their system? It would not necessarily conflict with the late stage consumption hypothesis.

Minor edits:
35: change "theropods" to "theropod"
125: change "m" to "mm"
135: put "the" in front of "marks"
143: the last part of the sentence is awkward. Please rephrase
155: change "examine" to "examined"
190: should it be "equidistances" (I'm honestly not sure)
192: changes "in" to "is"
248: change to "troodontid"
260: change "is" to "in" and "a" to "at"
262: change "makes" to "maker"
332: Change "great" to "greater" and "analogue" to "analogous"

---

## Round 0.2 · accepted · Accept

I confirm that your manuscript has been accepted for publication. Congratulations!

·

Basic reporting

I only found a couple of typos (a repeated "the" at line 114, and a "then" when it should be "than" at line 431). Otherwise, no comment.

Experimental design

No comment.

Validity of the findings

No comment

Additional comments

I appreciate the changes made, and this is a valuable contribution to the taphonomic literature. The authors have specified in their rebuttal that they would like to limit their mammal gnawing references to those where unguals/phalanges are modified. With this in mind, their citations are adequate.